

# Detecting the impact of land cover change on observed rainfall

Chun Xia Liang, Floris F. van Ogtrop and R. Willem Vervoort

School of Life and Environmental Sciences, Sydney Institute of Agriculture, The University of Sydney, New South Wales, Australia

## ABSTRACT

Analysis of observational data to pinpoint impact of land cover change on local rainfall is difficult due to multiple environmental factors that cannot be strictly controlled. In this study we use a statistical approach to identify the relationship between removal of tree cover and rainfall with data from best available sources for two large areas in Australia. Gridded rainfall data between 1979 and 2015 was used for the areas, while large scale (exogenous) effects were represented by mean rainfall across a much larger area and climatic indicators, such as Southern Oscillation Index and Indian Ocean Dipole. Both generalised additive modelling and step trend tests were used for the analysis. For a region in south central Queensland, the reported change in tree clearing between 2002–2005 did not result in strong statistically significant precipitation changes. On the other hand, results from a bushfire affected region on the border of New South Wales and Victoria suggest significant changes in the rainfall due to changes in tree cover. This indicates the method works better when an abrupt change in the data can be clearly identified. The results from the step trend test also mainly identified a positive relationship between the tree cover and the rainfall at $p < 0.1$ at the NSW/Victoria region. High rainfall variability and possible regrowth could have impacted the results in the Queensland region.

# INTRODUCTION

Land use and land cover changes can lead to changes in the local climate. Empirical and modelling studies have found that cloud types and rainfall are correlated to large scale vegetation cover changes, such as deforestation in the Amazon and in the Sahel (*Chagnon & Bras, 2005*; *Wang et al., 2009*; *Pinto et al., 2009*; *Mei & Wang, 2010*; *Kucharski, Zeng & Kalnay, 2013*; *Pitman & Lorenz, 2016*) and afforestation in south Israel (*Otterman et al., 1990*; *Ben-Gai et al., 1998*). Using airborne measurements in Western Australia, *Junkermann et al. (2009)* showed a significantly higher level of aerosols over an agricultural area compared to an adjacent area with natural vegetation. They suggested that a modification of aerosol concentrations due to deforestation could have contributed to a reduction of local rainfall, as more, but smaller rain droplets were observed. *Nair et al. (2011)* reported from the Bunny Fence Experiment in Western Australia that local land use change altered the synoptic west coast trough dynamics and surface roughness, and

Corresponding author
R. Willem Vervoort,
willem.vervoort@sydney.edu.au

this resulted in an observed rainfall decrease. Maximum temperatures were also found to be sensitive to land cover change in eastern Australia (*McAlpine et al., 2007*).

Overall the number of empirical studies analyzing changes to rainfall due to land cover change from observational data is limited. Most of the studies mentioned previously were either model simulations, or comparisons of modelled data with observations. This is because there are some fundamental experimental difficulties in both space (where does evaporated water reappear as rainfall?) and time (how much time does it take for land cover change effects to appear or disappear?). In addition, in many areas across the globe, rainfall variability is related to a complex set of interactions (outlined below), of which land use change might only be a minor component.

Locally, and on a shorter, daily time scale, there are two main sources that generate rainfall: moisture from advective atmospheric transport; and local evapotranspiration (*Eltahir & Bras, 1996*; *Bosilovich & Chern, 2006*; *Dirmeyer, Brubaker & DelSole, 2009*; *Gimeno et al., 2010*). The local evapotranspiration component is the component considered to be affected by land use change (*Eltahir & Bras, 1996*). According to *Trenberth (1999)*, the contribution of advective moisture partially depends on the availability of external moisture and atmospheric transport. On the longer time scale, such as monthly and annually, large scale atmospheric dynamics are affected by large scale climate drivers. For example, many studies have reported significant relationships between rainfall in large parts of Australia and the El Niño-Southern Oscillation (ENSO) (*Verdon et al., 2004*; *Risbey et al., 2009*; *Speer, Leslie & Fierro, 2011*). In contrast, local evapotranspiration (ET) is determined by local land surface characteristics, which influence local scale atmospheric dynamics and hence the amount of rainfall, including contribution from both main sources.

Although climate drivers demonstrate some capability to predict Australian rainfall, there is still a large amount of unexplained variance. *Westra & Sharma (2010)* pointed out that models based on global sea surface temperature anomalies can only predict up to 14.7% of annual precipitation variance. More generally, some of the remaining variance could be due to land surface processes as suggested in studies predicting local rainfall (e.g., *Ma et al., 2011*; *Zeng et al., 2012*; *Pitman & Lorenz, 2016*; *Saha, Dirmeyer & Chase, 2016*). However, most are based on modelling experiments and few empirical observational studies have been reported. *Pitman et al. (2004)* found a good match between observations and simulated rainfall changes in southwest Western Australia, forced by land cover change. *Timbal & Arblaster (2006)* were able to reproduce the rainfall decline in south west Australia by including land cover influence. In addition, local land use change might not be a primary, but is likely to be a secondary cause of rainfall change (*Nicholls, 2006*).

Therefore, the aim of this study is to use a statistical approach on rainfall data at regional scales to investigate the cause and effect relationship between land cover change and local rainfall, which is demonstrated in many modelling studies. More specifically, we hypothesize that a step change in land cover on the surface will cause a step change in the rainfall. To demonstrate this we study changes in observed rainfall over time at a Queensland and a NSW/Victoria location where there are possible step changes in land cover change due to land clearing and bush fires. The methodology uses statistical

approaches to identify changes in rainfall, which are subsequently associated with land cover change through spatial comparison.

In this paper, after this section (the introduction), 'Study regions and tree cover change' covers the case study areas and the observed land use change. 'Data' describes the data used in the study in more detail. 'Statistical method' details the statistical methods and the underlying assumptions related to the modelling approach, 'Results' gives the results, which are further discussed in 'Discussion' and finally 'Conclusions' offers the conclusions.

## Study regions and tree cover change

In Australia, significant tree cover change has mainly occurred in the north east and south east of the continent, as well as in the southwest of Western Australia. According to the National Dynamic Land Cover Dataset (DLCD) (*Lymburner et al., 2010*), most of these areas have experienced decreases in the Enhanced Vegetation Index (EVI) post 2000, as derived from satellite data. As an index for vegetation greenness, the decreasing values of EVI indicate lower biomass over time in the tree cover regions. The possible EVI reduction might be due to land clearing, bush fires or drought.

Two regions were selected where significant tree cover change since 2000 was reported. The first region is located in south central Queensland (QLD) partly covering the north of the Murray Darling Basin (MDB) (site 1 in Fig. 1). High rates of land clearing have been reported in this region during the early 2000s (*Department of Natural Resources and Water, 2007*). The second study region is located at the border of New South Wales and Victoria (NSW/VIC), and includes the Snowy Mountain ranges (site 2 in Fig. 1). Severe bush fires occurred in this area in early 2003 (see Fig. 2). The 2003 bush fires were the largest in the last 60 years (*The State Government of Victoria, 2011*). Two thirds of Kosciuszko national park was heavily burned and regrowth was suggested to be slow due to drought and cold conditions (*ABC News, 2003*), and the type of species in this region. However, in the longer term, after an early high transpiration period a recovery of pre-fire (ET) would be expected (*Kuczera, 1987*). For the purpose of this study, significant tree cover loss has happened in both study areas in the last decade, either permanently or temporarily.

The two regions have different characteristics. The QLD region is partially grassland and subtropical, while the NSW/VIC region is mainly within the temperate zone, under the Köppen classification. According to Australian Bureau of Meteorology (BoM), the NSW/VIC region receives 1000–2000 mm rainfall annually, which is more than double the annual rainfall in the QLD region. Evapotranspiration is similar in both regions. Marine moisture and orographic effects are likely to be the main contributors to rainfall in the southeast mountain areas of the NSW/VIC region.

The land use and land cover characteristics in the two regions are also different. In the Queensland region, the tree cover is sparse over most of the area. The MODIS satellite tree cover data (discussed in more detail in 'Data') shows that tree cover in this region is generally below 20% of total ground area. Grazing is the main activity in this region, with over 90% of land used by the grazing industry (*ABARES, 2010*). Our starting assumption is that the main cause of the EVI decline over large part of the region is due to land clearing.

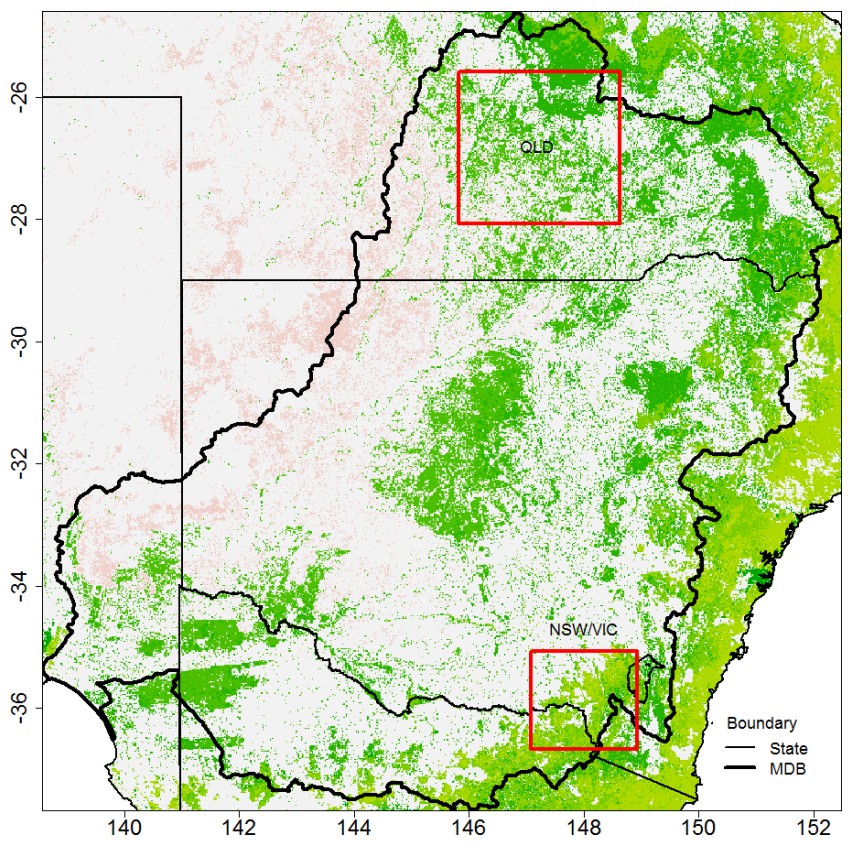

**Figure 1** **Selected study regions are highlighted by red rectangles in the main map.** The types of tree cover in 2008 from the DLCD product is shown at the background. In site 1 (the QLD region), the tree cover is mostly sparse. In site 2 (the NSW/VIC region), many areas have open or close forest where the tree cover is denser.

Tree cover has been cleared at a massive scale over the last decade, especially during 2002–2004. The reports from the Queensland Statewide Land Cover and Trees Study (SLATS) (e.g., *Department of Natural Resources and Mine, 2005*; *Department of Science, Information Technology and Innovation, 2017*) were used to investigate the time and location of the land clearing in the QLD region.

The Kosciuszko national park is within the NSW/VIC region. Here tree cover is denser with open or even closed forest (the tree cover distribution is bimodal at 10–20% and 60–70%). The dominant species in the alpine area are Snow Gum (*Eucalyptus Pauciflora*)and large stand species such as Alpine Ash (*Eucalyptus delegatensis*) and Mountain Gum (*Eucalyptus dalrympleana*) in the sub-alpine area. These trees can reach a great height but they take long time to grow. For example, Alpine Ash (*Eucalyptus delegatensis*) would need about 20 years to mature (40–45 m, *Buckley et al. (2012)*). This region is vulnerable to fires and drought; however land clearing is not a major issue. The MODIS burned area product, MCD45A1 (*Roy, Lewis & Justice, 2002*; *Roy et al., 2005*; *Roy et al., 2008*), was used to locate bush fires areas in the NSW/VIC region, with a grid resolution of 500 m. MCD45A1

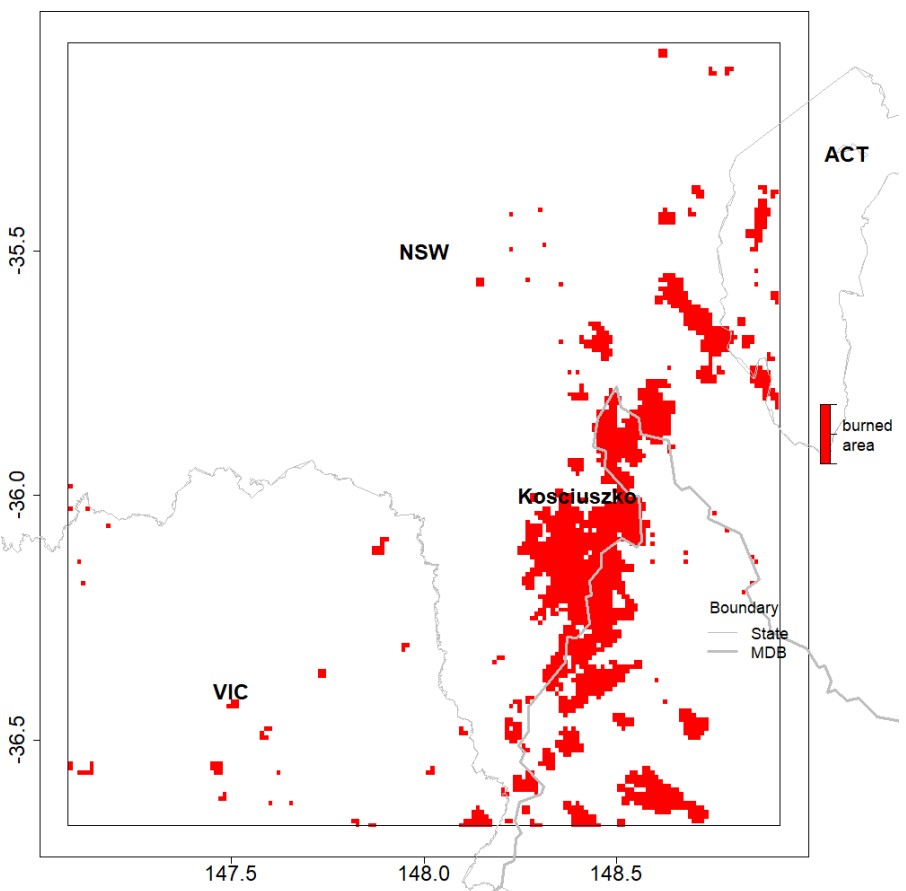

**Figure 2** **Location of bushfires occurring in January 2003, in and around the NSW/VIC study region, as shown by the red pixels.** The map shows the large area in the Kosciuszko national park that has been burned. Some locations in the southwest of the Australian Capital Territory (ACT) have also experienced intensive bushfires.

provides monthly burning information on all pixels, which helps to pinpoint an abrupt event.

Due to the difference in nature of the land cover change in the two regions, the post-change vegetation status is hypothesized to be different as well (see Fig. 3). The overall hypothesis is that the effect of 2003–2004 land clearings in the QLD region and the 2003 bush fires in the NSW/VIC region cause a step change in the local rainfall. The actual tree cover change during this time at the pixel level was derived from the 15-year MODIS data (discussed below). As the length of the tree cover data is shorter than available rainfall data, earlier land clearing in the QLD region cannot be identified spatially, hence they are excluded from the analysis.

## DATA

Several land surface data sets were used in this study. The main one was the MOD44B product Global Vegetation Continuous Field data set (version 5). This data set provides
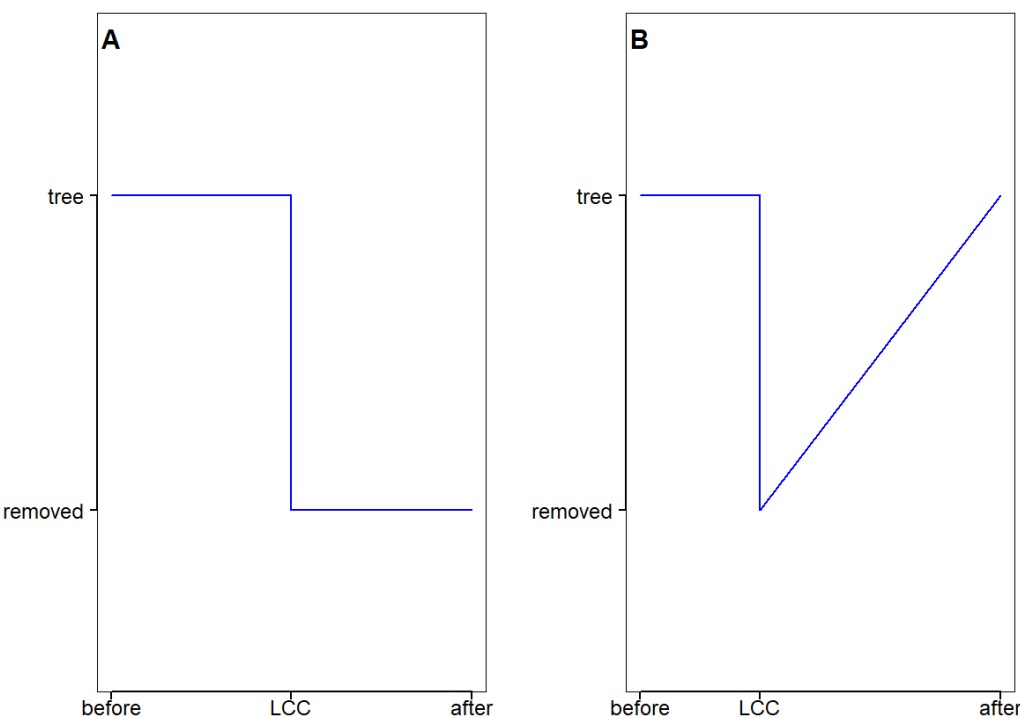

**Figure 3** The expected evolution of the land surface after trees have been removed in (A) the QLD region and (B) the NSW/VIC region.

estimates of percent tree cover (percentage of ground surface covered by trees) at a grid resolution of 250 m (*Townshend et al., 2011*), which is finer than the earlier mentioned burned product MCD45A1. The data set is available on an annual time interval for the study period of 2000–2015. The tree cover data was produced from 16-day Terra MODIS Land Surface Reflectance data and Land Surface Temperature (*Townshend et al., 2011*). The National Dynamic Land Cover Dataset (DLCD) (*Lymburner et al., 2010*) from the Australian Collaborative Land Use Mapping Program (ACLUMP) was used to verify the trend of vegetation cover change calculated from the previous data set. This data set, developed by Geoscience Australia and Australian Bureau of Agricultural and Resource Economics and Sciences (ABARES), is the first nationally consistent and thematically comprehensive land cover reference for Australia. The DLCD is based on the 16-day Enhanced Vegetation Index (EVI), again from the MODIS satellite, between April 2000 and April 2015. It also has a grid resolution of 250 m. The data set provides information on the final land cover types (as in 2015) and estimated trend of EVI statistics (annual mean, maximum and minimum).

The SILO Rainfall product data for Australia was used (*Jeffrey et al., 2001*) (available online at https://silo.longpaddock.qld.gov.au/). The data has been projected onto a national 0.05° by 0.05° grid (approximately five km by five km). This gridded data set was generated from station observations using spline interpolation and kriging (*Jeffrey et al., 2001*). The data has been compared to other gridded products and observed data and is generally of

high quality (*Tozer, Kiem & Verdon-Kidd, 2009*; *Tozer, Kiem & Verdon-Kidd, 2012*). The data is available on a daily and monthly basis from 1889 to current. Here a subset of 36 years (1979–2015) was used. The study was conducted on monthly data, as a land cover change effect on annual rainfall might be negligible, but can often be significant in particular months or seasons (e.g., *Otterman et al., 1990*; *Gaertner et al., 2001*; *Semazzi & Song, 2001*; *Oleson et al., 2004*; *Deo et al., 2009*).

Large scale climate drivers are represented by various climatic indices. The Southern Oscillation Index (SOI) is generally regarded as a good predictor of Australian rainfall (*Risbey et al., 2009*; *Chowdhury & Beecham, 2010*; *Westra & Sharma, 2010*), but its skill is weaker in some parts of Australia. For example the Southern Annular Mode (SAM) is found to be more important than ENSO in south Western Australia (*Meneghini, Simmonds & Smith, 2007*). The testing of the suitability of each index for the regions of interest is described in a later section. The following climate indices were used as candidate predictors for local rainfall.

- Southern Oscillation Index (SOI). The Troup version of the monthly SOI series used in this study was obtained from BoM (available online at http://www.bom.gov.au/climate/current/soihtm1.shtml).
- Eastern, East Central and Central Tropical Pacific Sea Surface Temperatures (NINO 3, NINO 3.4 and NINO 4). Monthly SST anomalies are available from IRI/LDEO data library and the extended NINO data set is used (available online at http://iridl.ldeo.columbia.edu/SOURCES/.Indices/.nino/.EXTENDED/).
- Pacific Decadal Oscillation (PDO). The Pacific Decadal Oscillation is the leading principal component of monthly SST anomaly in the North Pacific Ocean.. The monthly PDO series was provided by JISAO (Joint Institute for the Study of the Atmosphere and Ocean, University of Washington) (available online at http://jisao.washington.edu/pdo/PDO.latest).
- Indian Ocean Dipole (IOD). The Indian Ocean dipole is commonly measured by the difference between SST anomaly in the western (50–70°E and 10°S–10°N) and eastern (90–110°E and 0–10°S) equatorial India Ocean (*Saji et al., 1999*). Monthly IOD was obtained from JAMSTEC (the Japan Agency for Marine-Earth Science and Technology) (available online at http://www.jamstec.go.jp/frcgc/research/d1/iod/DATA/dmi.monthly.txt).

## STATISTICAL METHOD

As an initial analysis, a simple boxplot and $t$-test is used to analyse whether there is a significant change in tree cover in time, especially before and after the suspected change in the regions.

To assess the actual causal relationship between the tree cover and the rainfall, a flexible regression model is applied (discussed in detail below). A step change is not directly obvious in the time series of the rainfall anomalies (Fig. 4) for both regions, even though the data is deseasonalised and detrended. In this study we apply different statistical methods to analyse the effect of tree cover change on rainfall. Both methods make use of a regression
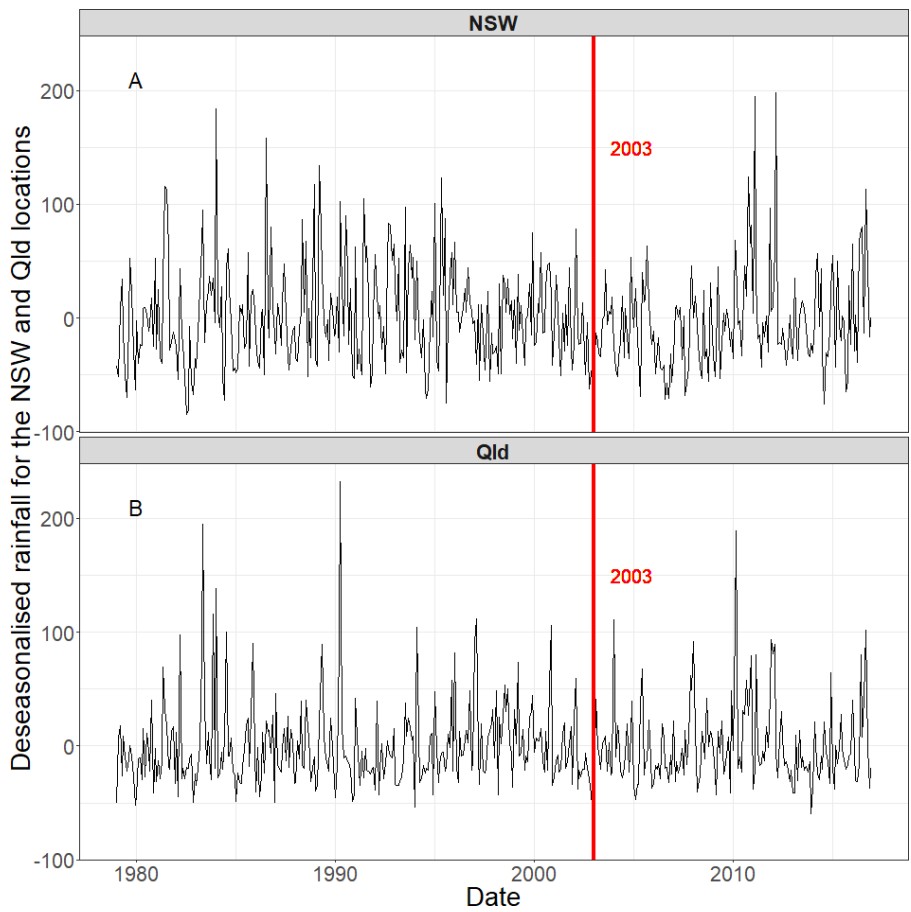

**Figure 4** **The deseasonalised and detrended rainfall over the 30 years period in (A) the QLD region and (B) the NSW/VIC region.** The vertical red lines indicate the year of 2003, in which the studied land cover changes occurred. A change in the time series data is not obvious before and after the land cover changes.

model to remove year-on-year variability in rainfall to strengthen the tree cover change signal.

In the first method, the tree cover change is implemented as a factor variable in the regression model, and the significance of this variable is tested. In the second method, a rank sum test (step trend test), is applied to the regression model residuals, after effects of other major factors were removed. This assumes that after removal of all climate influence, long term linear trend and seasonal variation, the vegetation cover change is the only factor explaining the non-random pattern in the rainfall residuals.

## Regression model

As highlighted in the introduction, the Australian climate is influenced by sea surface temperatures in the tropical Pacific and Indian Oceans, as well as pressure systems in the Southern Ocean (*BoM, 2012*). *Risbey et al. (2009)* compared five large-scale drivers, including ENSO (measured by SOI and the Tropical Pacific Sea surface temperatures (SSTs)), IOD, SAM, MJO (Madden-Julian oscillation) and blocking, in relation to

Australian rainfall variability. The MJO is a large scale eastward-propagating wave-like disturbance located around equatorial latitudes (*Risbey et al., 2009*). They identified SOI as the most important index among all climate indices tested for broad parts of Australia (including QLD and NSW/VIC) in almost any season. In this study, four climate indices were selected from the main climatic indicators (see section 'Data') and used as the explanatory variables in the model for each study region. A further complicating factor is the influence of the ''millenium drought'' over the study period and in particular the change to wet conditions in 2010–2011 (*Van Dijk et al., 2013*). Therefore, the spatially averaged monthly rainfall in the Murray Darling Basin (MDB, downloaded from the Bureau of Meteorology) was used to explain the year-on-year variation in the rainfall in the regions. Since both regions at least partly overlap with the MDB, the average rainfall for the entire basin was assumed to be a useful explaining variable.

Correlations between rainfall and each climate index were analysed. Rainfall in each study region was first deseasonalised and detrended using the seasonal decomposition function ds in the package deseasonalise in R (*R Core Team, 2018*). Using detrended data gives a better indication of the underlying correlation (*Smith & Timbal, 2012*). The cross-correlations between the deseasonalised and detrended rainfall and the climatic indices were tested, and the strongest indicators at lag 0 were identified for the model. Because the PDO describes the multi-decadal SST with lower frequency (*MacDonald & Case, 2005*; *Zanchettin et al., 2008*; *Kamruzzaman, Beecham & Metcalfe, 2011*), instead of 37-year rainfall data, a longer period (115 years, from 1900 to 2015) was used to estimate the correlation with PDO, up to lag 24. For the other indices, the 37-year data was used.

Rainfall in Australia shows strong seasonal patterns (*Holper, 2011*; *Australian Bureau of Statistics, 2012*). As a result, a seasonal component of rainfall has a periodic pattern which should be included in the model. In addition, long term trends in the regional rainfall in some parts of Australia are significant (*Hughes, 2003*; *Gallant, Hennessy & Risbey, 2007*; *Chowdhury & Beecham, 2010*). The presence of long term trends can be confused with the outcome of a step change in rainfall. As a result a linear trend term was implemented in the model to remove any long term effects.

We assumed all the factors are additive smooth components in determining rainfall following *Kamruzzaman, Beecham & Metcalfe (2011)*. In this case, the rainfall model is a generalised additive model (GAM) (*Hastie & Tibshirani, 1986*) with a log link function g() and assuming the residuals are gamma distributed. We used the shrinkage version of the cubic regression splines (*Wood, 2011*) as smooth function. All splines were limited to 3 knots in flexibility (*Wood, 2011*), to reduce the risk of overfitting.

$$g(E(\mathbf{R}_r)) = \begin{aligned} &\beta_0 + s_1(\mathbf{MDB}_{\text{monthlyRain}}) + s_2(\mathbf{SOI}) + s_3(\mathbf{IOD}) + \\ &s_4(\mathbf{Nino3.4}) + s_5(\mathbf{Nino4}) + s_6(\mathbf{PDO}) + \\ &s_7(\mathbf{Season}) + \beta_1 \mathbf{Trend} + \epsilon_r \end{aligned} \tag{1}$$

The bold letters represent the time series vectors. The region is indicated by $r$, while $\beta_u$ ($u = 0, 1$) are the fitted coefficients in the model. $s_v$ ($v = 1, 2, 3, \ldots$) are the smooth penalized cubic regression spline functions on the climatic indices and the season.

$MDB_{monthlyRain}$ is the spatially averaged monthly rainfall across the Murray Darling Basin.

The variable **Trend** $= 1, 2, 3 \ldots n$, where n is the total number of months in the time series is the possible long term trend in the data, while **Season** is the seasonal component. The climatic terms are also modelled with smooth functions as the effect of large scale drivers on Australian rainfall can be seasonal (*Murphy & Timbal, 2008*; *Schepen, Wang & Robertson, 2012*).

### Tree cover change as factor variable

One of the main difficulties in empirical observation studies related to the effect of land cover change on rainfall is the lack of continuous monitoring of land surface variables. Moreover, no specific variable can possibly be defined that can clearly represent the land surface process. Given the lack of a full picture of the land surface process, a factor variable was used in the regression model to represent the abrupt land surface change. The change could be a result of either land clearing or bush fires as long as it is permanent or takes a long time to recover. As indicated we approached the problem in two different models ways.

In the first method, the tree cover change was used as a predictor in the regression model, represented by a factor variable **LC**.

$$LC = \begin{cases} \text{Trees} \\ \text{Removed} \end{cases} \tag{2}$$

Therefore in both regions, land cover is "trees" for the period before land cover change and "removed" for the period after the change. Here we simply assumed that vegetation cover change has occurred on every pixel. The remaining term $\epsilon_r$ is the amount of rainfall that is attributed to other unspecified factors and random errors. Hence the regression model becomes

$$
\begin{aligned}
g(E(\mathbf{R}_r)) = \quad & \beta_0' + s_1'(\mathbf{MDB_{monthlyRain}}) + \\
& s_2'(\mathbf{SOI}) + s_3'(\mathbf{IOD}) + \\
& s_4'(\mathbf{Nino3.4}) + s_5'(\mathbf{Nino4}) + s_6'\mathbf{PDO} + s_7'\mathbf{Season} + \\
& \beta_1'\mathbf{Trend} + \beta_2'\mathbf{LC} + \epsilon_r'
\end{aligned}
\tag{3}
$$

One of the difficulties is to point an exact time to the changes in the vegetation cover in the two regions. In the QLD region, no exact time can be assigned to the land clearing. According to the SLATS reports, the most substantial clearing occurred between 2003–2004. However, the information on the change in type of land cover during the time period is missing. Four scenarios were initially tested in the analysis, however there was no real difference between these scenarios. In the NSW/VIC region, severe bush fires were reported in early January 2003. Hence the "tree" cover state was up to December 2002 then it was changed to "removed" state from January 2003. As a starting date, the regression model was run from 1979 for both regions.

### Step trend test

To support the regression analysis, a Mann–Whitney Rank-Sum step trend test was used to detect changes in rainfall as a result of vegetation cover change. This specific nonparametric
**Table 1 Example of ranking rainfall residuals.**

| Year | Rainfall residuals | Rank $R'_{1k}$ |
|------|--------------------|----------------|
| 1998 | −0.3  | 6 |
| 1999 | −60.9 | 2 |
| 2000 | −16.1 | 4 |
| 2001 | −71.7 | 1 |
| 2002 | 111.1 | 7 |
| 2005 | −7.2  | 5 |
| 2006 | −60.5 | 3 |

statistical test was modified from the Mann–Whitney $U$ test by *Hirsch & Gilroy (1985)* and can identify a step change in data which is cross-correlated. In this case, this is important as the gridded rainfall dataset has a high spatial correlation between neighbouring pixels. The advantages of using the Rank-Sum test are: (1) it does not depend on assumptions of the data distribution; (2) it is not restricted to datasets with no missing data; (3) it is robust and not as easily influenced by outliers and negative numbers (*Hirsch & Gilroy, 1985*). However, the test has less power than parametric tests. As a nonparametric rank-based test, it depends on the ranks of the data.

The rainfall residuals from the regression model in Eq. (3) were used. The assumption is that the regression model deseasonalises and detrends the rainfall data (*Hirsch & Gilroy, 1985*), amplifying the local landuse effects. For each month, rainfall residuals of each year were ranked in an ascending order. The ranking of January rainfall in a sample pixel $k$ in QLD is illustrated in Table 1.

The before and after period in the data formed two groups of samples. The split point of the two periods was based on the timing of the vegetation cover changes. In the QLD region, changes occurred anytime during 2003 and 2004. In contrast to the previous method, the time period covering the land cover change was excluded, as the nonparametric test allows missing data and the power of the test is greater if data of the change period is ignored (*Hirsch & Gilroy, 1985*). As a result, the post change period was 2005–2015 for the Queensland location. In the case of NSW/VIC, the bushfires broke out in early January 2003. The change was within a relatively short period of the year. Therefore the post change period in this region still started in January 2003. The pre-change period was set to five years (1998–2002) in both regions.

The rank of rainfall in month $j$ year $i$ in pixel $k$ is denoted as $R'_{ijk}$. The sum of ranks of rainfall in month $j$ in pixel $k$ before the known intervention is:

$$W_{jk} = \sum_{i=1}^{n_1} R'_{ijk}. \tag{4}$$

$n_1$ is the number of years before the land cover change. The expected value of $W_{jk}$ is

$$\mu_w = n_1(n_1 + n_2 + 1)/2 \tag{5}$$

$n_2$ is the number of years after the change. Hence the expected value of the rank sum before the intervention is the same for all months and all pixels. The sum of ranks for

the whole time period is fixed, as $(n_1 + n_2)(n_1 + n_2 + 1)/2$. In this study, since there are only two groups (before and after), knowing the rank-sum of one group is the same as knowing the rank-sum of the other group. If the rainfall data is temporally and spatially independent, the variance of $W_{jk}$ is

$$\sigma_w^2 = n_1 \cdot n_2(n_1 + n_2 + 1)/m \tag{6}$$

where $m$ is the number of months which is 12 in the case of a full year.

Here the deseasonalised and detrended data shows little autocorrelation in time, but possesses strong cross-correlation between neighbouring pixels, i.e., $R > 0.99$.

The sum of $W_{jk}$ for a block of $ns$ pixels over the whole year, $\sum_{j=1}^{12}\sum_{k=1}^{ns} W_{jk}$, has mean

$$E(\sum_{j=1}^{12}\sum_{k=1}^{ns} W_{jk}) = 12 \cdot ns \cdot \mu_W \tag{7}$$

and variance

$$Var(\sum_{j=1}^{12}\sum_{k=1}^{ns} W_{jk}) = \sum_{j=1}^{12}\sum_{k=1}^{ns}\sum_{h=1}^{ns} C(W_{jk}, W_{jh}). \tag{8}$$

$C(W_{jk}, W_{jh})$ is the covariance of the $W$ statistics between pixel $k$ and pixel $h$ in month $j$. When $k = h$, $C(W_{jk}, W_{jh}) = \sigma_w^2$. When $k \neq h$,

$$C(W_{jk}, W_{jh}) = \sigma_w^2 r(R_k, R_h) \tag{9}$$

where $r(R_k, R_h)$ is the product moment correlation coefficient of the concurrent ranks in pixel $k$ and $h$. Here $r$ is calculated on the full time series in each pixel. In the analysis, the test was applied to a square block of four pixels each time. As argued by *Hirsch & Gilroy (1985)*, $ns = 4$ is the most optimal solution to balance the cost and the gain in the test power.

The statistic of the step trend test is then defined as

$$Z' = \frac{\sum_{j=1}^{12}\sum_{k=1}^{ns} W_{jk} - 12 \cdot ns \cdot \mu_w}{\sqrt{Var(\sum_{j=1}^{12}\sum_{k=1}^{ns} W_{jk})}}. \tag{10}$$

The above statistic is written for a 12 month period. By changing the value 12, it can also be used to test seasonal rainfall change or for other customized periods.

The null hypothesis ($H_0$) in this study is that there was no change in rainfall due to land surface intervention. The results of the step trend test can be interpreted according to the sign of the $Z'$ score (see Table 2, Chapter 23, P887 (*Hipel & McLeod, 1994*)), and is normally distributed similar to the standard normal statistics Z.

As part of the analysis, the "field significance" of the $Z'$ score test was considered to improve the interpretation of the step change at regional scales from multiple local tests (*Wilks, 2006*; *Westra, Alexander & Zwiers, 2013*). Here, the bootstrapping resampling method from *Westra, Alexander & Zwiers (2013)* was used to evaluate the field significance. This means the spatial structure of the pixels was maintained, but the order of the years and months was changed by random resampling. For each resampling, the test statistic

**Table 2** **The interpretation of $Z'$ score in the step trend test.**

$Z' > 0$ and rainfall decreases after change

$Z' < 0$ and rainfall increases post change

$Z' = 0$ and rainfall does not change

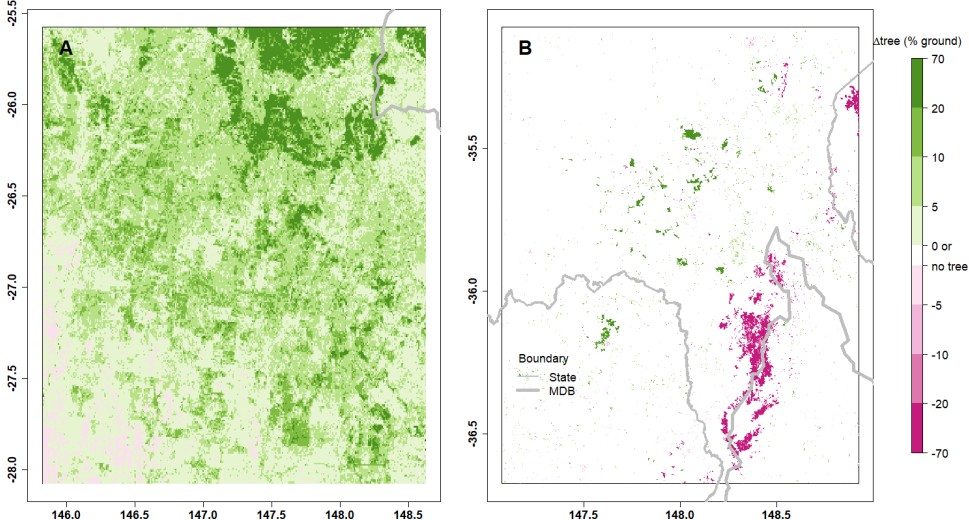

**Figure 5** **The maps show significant changes in tree cover identified from the MOD44B data between 2003 and 2015 in the Qld region (A) and the NSW/VIC region (B).** The amount of change was calculated as the difference in tree cover before and after the specified land cover intervention and it is shown as the percentage of the ground area. Green colour indicates an increase in tree cover, while red colour indicates a decrease in tree cover.

identifies the percentage of the pixels with a significant positive or negative step change for the step trend test. The test statistics on 1,000 resampled replicates were used to develop the distribution of these percentage values.

# RESULTS

## Tree cover change

The pixels, where the tree cover change based on the MOD44B data was significant ($p \leq 0.05$) in each study region, are shown in Fig. 5 for the QLD region (Fig. 5A) and the NSW/VIC region (Fig. 5B). In the QLD region there actually has generally been an increase in tree cover after the clearing of native vegetation stopped. In the NSW/VIC region, much of the tree loss between 2002 and 2003 was concentrated in the Snowy Mountains close to the border of NSW and VIC, which is also evident in the figure. Tree cover loss occurred in large parts of the QLD region between 2002 and 2005, but this tree loss was spatially less concentrated.
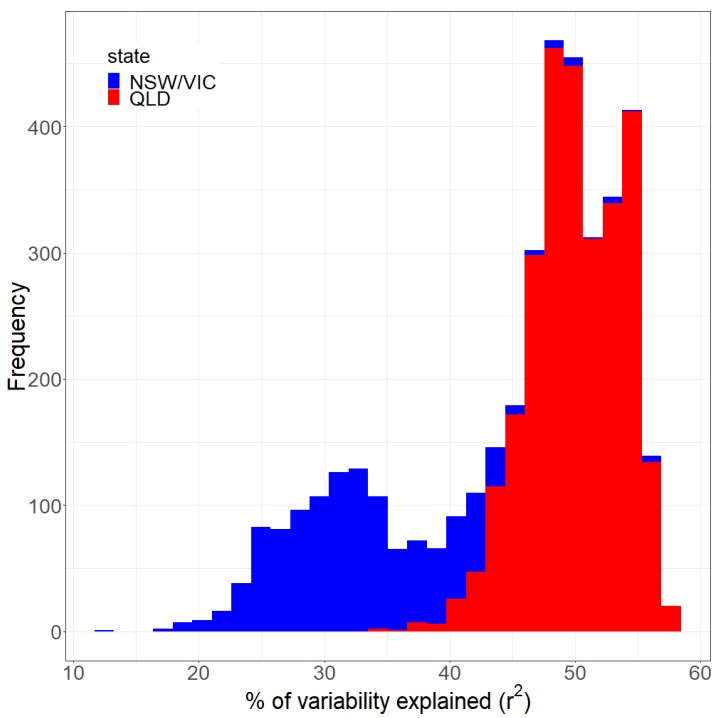

**Figure 6** The distribution of the variance explained (adjusted $r^2$) by the full regression model.

## Regression model

The correlation between the climatic indices and rainfall in the regions (See the Supplemental Information with this paper) indicated that for both regions the PDO had the weakest correlations, and this factor was therefore dropped as a predictor. Although some indices were serially correlated with rainfall up to several months, the lag zero events had the greatest correlation coefficients. Furthermore, using multiple climatic index series was generally found most useful in rainfall prediction (e.g., *Risbey et al., 2009*; *Kamruzzaman, Beecham & Metcalfe, 2011*).

Overall the regression model only explains a limited amount of the rainfall variability (Fig. 6). The model in Eq. (3) accounts for an average around 33% of the variation in the NSW/VIC region and about 50% of the variation in the QLD region as indicated by the adjusted $r^2$. The adjusted $r^2$ is the coefficient of determination, a measurement of the amount of variability predicted by the model adjusting for the number of explanatory terms.

In terms of predictors of the local rainfall in the model, logically, the average rainfall across the larger Murray Darling Basin is highly significant. This confirms that this variable is a good reflection of the year on year variability in the rainfall. The model also confirms the importance of the climate drivers and the seasonality in Australian rainfall, as many of these variables were significant. Even at the grid level, the seasons and several of the climatic indices were significant ($p \leq 0.05$) everywhere in both regions. The climate drivers (at lag zero) accounted, on average, for 6.5% of the rainfall variability in both the QLD
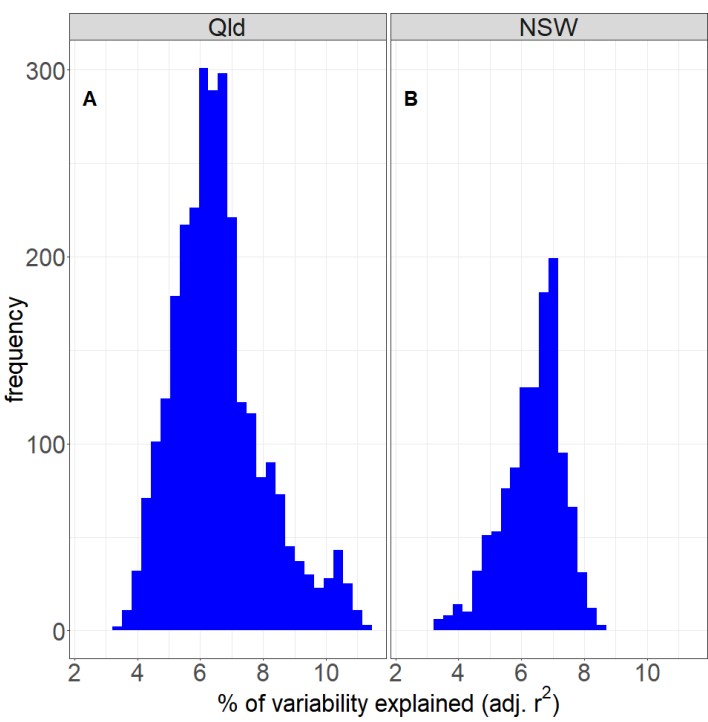

**Figure 7** **The performance of the regression model if rainfall is only modelled by the climate drivers.** It shows the percentage of rainfall variability that can be explained by the climate drivers for the Qld (A) and NSW/VIC (B) region.

region and the NSW/VIC region (see Fig. 7 for the distribution of adjusted $r^2$ in these two regions). These figures are within the upper bound of seasonal rainfall predictability by a SST anomaly field reported by *Westra & Sharma (2010)*.

There were generally no statistically significant long term trends in both regions. However, the trend term was kept in the regression model to ensure the detection of step change was not due to a possible long term trend (even if this was not significant).

The land cover variable in the model (Eq. (3)) aims to identify a step change in the rainfall before and after the observed change in land cover. The variable was mainly significant ($p \leq 0.05$) for the rainfall estimates in some areas in NSW/VIC, as shown in Fig. 8B. However, the number of pixels where the landcover variable was significant was much greater in NSW/VIC compared to the number in the QLD region (Fig. 8A). There was also some relationship between the areas of bushfires in Fig. 2. Only a very small area with a significant step change due to the land cover changes was found in rainfall in the QLD region (Fig. 8A).

More generally, the model for NSW/VIC suggests that the land cover variable has a positive impact on rainfall in both regions. The fitted coefficients for the Land cover change variable were consistently positive for the "tree" part of the series. It implies that rainfall was higher when the surface was covered by trees.

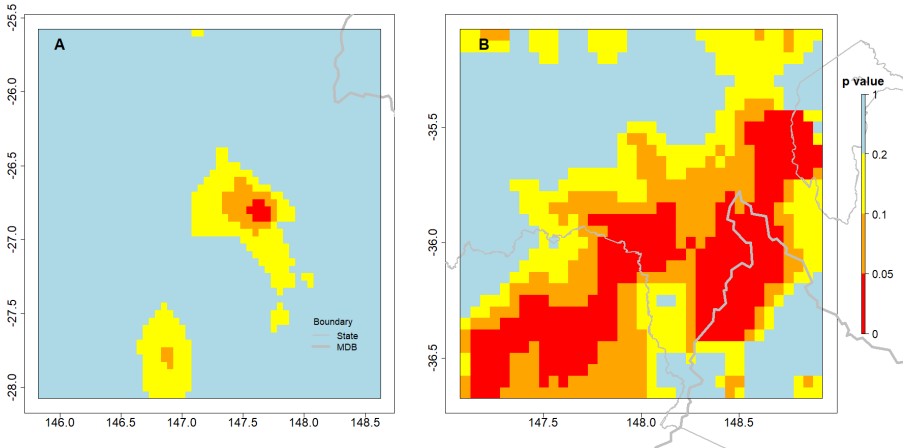

**Figure 8** The spatial distribution of significance of land cover step change variable in the GAM model predicting changes in rainfall in both regions, the QLD region (A) and the NSW/VIC region (B). The outlines of relevant Australian states and the Murray Darling Basin are indicated in grey lines. The *p*-value reported is for the land cover variable in the model.

## Stronger step trend in NSW/VIC compared to QLD

In both regions there are areas of positive $Z'$ values which imply a decrease in rainfall (Fig. 9), but this is stronger in the NSW/VIC area (Fig. 9B) than in the QLD area (Fig. 9A). Qualitatively the locations where changes in tree cover occurred in the NSW/VIC area (Fig. 5B) seem to agree with the patterns in the $Z'$ scores. However, a direct relationship would not necesssarily be expected as movement of air masses could mean that actual changes of rainfall are observed close by, but not necessarily exactly at areas with changes of landcover. For the QLD region (Fig. 9A) there are only a few significant $Z'$ scores which matches the earlier significance in the landcover variable in the full regresion. In the QLD region, only 0.2% of the pixels obtained a positive $Z'$ score with $p < 0.1$. In the NSW/VIC region (Fig. 9B) 3.7% of pixels have a positive $Z'$ score with $p < 0.1$. In general it is only a small proportion of both study regions.

The rainfall regression residuals without the landcover variable for the two regions (before-change since 1979 and after-change) were also compared using a simple *t*-test. The mean rainfall residuals were significantly different ($p < 0.05$) between the "before" and "after" periods in both regions. For the Queensland locations, the mean monthly rainfall residuals were slightly lower ($p < 0.05$) after the change in landcover ($\sim 0.45$ mm/month). For the NSW/VIC locations there was larger decrease in the mean monthly rainfall residual ($p < 0.05$) post change ($\sim 1.5$ mm/month).

The observed fraction of significant $Z'$ scores (at $p < 0.1$) is plotted on the results of the bootstrap distribution (Fig. 10). The results support the earlier $Z'$ score results. Only the percentage significant positive $Z'$ scores, indicating a decrease in rainfall for the NSW/VIC region as a result of a decrease in tree cover, falls on the tail of the field significance distribution, while all the others are well within the distribution. This suggests that the percentage of significant positive $Z'$ scores in the NSW/VIC region is least likely due to

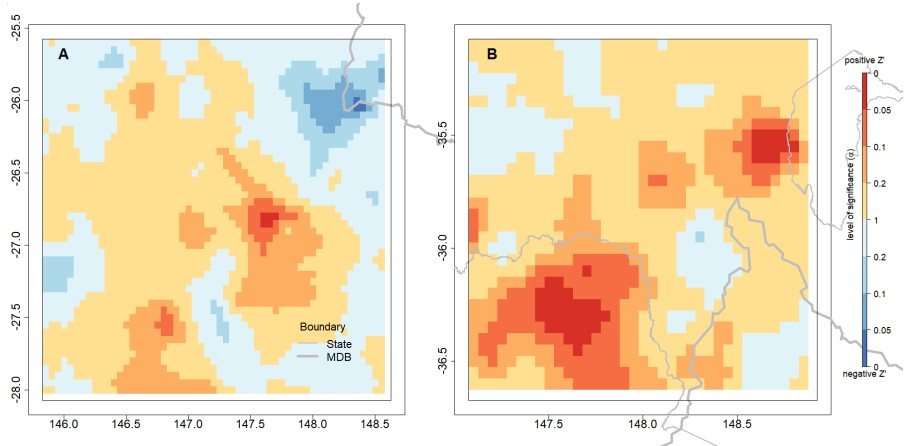

**Figure 9** Spatial distribution of the step trend test $Z'$ statistics in the two study sites, the QLD region (A) and the NSW/VIC region (B). Warm colours (yellow, orange and red) are for positive $Z'$ values which indicate decreasing rainfall trend due to the land surface change. Cold colours (light blue to blue) are for negative $Z'$ values which indicate increasing rainfall trend. The deeper the colour, the more significant the statistic.

a unique series of rainfall years, but is most likely due to changes in landcover. In other words, in the NSW/VIC region it is most likely that the change in landcover (decrease in tree cover) caused a decrease in the rainfall.

## DISCUSSION

Overall the summary table (Table 3) of the results indicates that the effects of land clearing or bushfire on rainfall are consistent across the range of different statistical tests. In all cases, the data for the NSW/VIC region indicates statistical evidence that the reduction in tree cover (the land cover change) resulted in a local decrease in rainfall. In contrast, the data for the QLD region indicates lower statistical evidence that the loss of trees due to land clearing resulted in reduction in the rainfall (Table 3).

Generally, empirical studies on LCC-precipitation interaction are conducted within an area with known land surface intervention (e.g., *Otterman et al., 1990*; *Durieux, Machado & Laurent, 2003*; *Negri et al., 2004*; *Sato, Kimura & Hasegawa, 2007*). However, these locations are rare and difficult to isolate from real landscape change. Modelling studies are abundant (e.g., *Chagnon & Bras, 2005*; *Wang et al., 2009*; *Pinto et al., 2009*), but these are generally not directly linked to observed data. In this study we tested the effect of land cover change across a broad region, which included locations where changes were known to occur or have occured. The advantage of the current approach is that long time series of land cover data are not required. Furthermore, it does not assume a specific relationship between vegetation cover change and rainfall but allows the data to show this relationship, by applying the analysis to a broader area outside the boundary of the vegetation cover change. This approach is expected to provide a way to reduce the risk of a false positive paradox, by comparing results between areas with and without vegetation cover change.
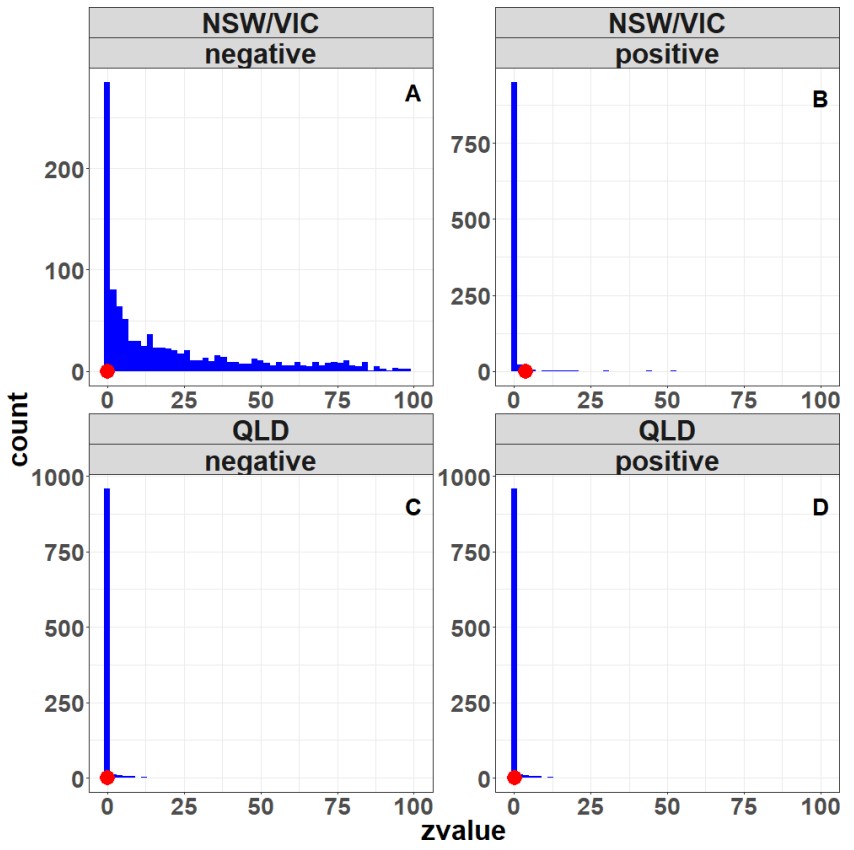

**Figure 10** Field significance results showing the distribution of percentage of significant $Z'$ scores for the resampled series (blue bars), and the precentage significant $Z'$ scores for the original series of rainfall years (red circle).

**Table 3  Summary table of all tests on the two regions.**

| Test | QLD | NSW/VIC |
|---|---|---|
| LC variable | Small area of significant pixels in the centre | Large area possibly aligning with bushfire affected area. |
| $t$-test regression model residuals | Slightly lower mean residual ($p < 0.05$) after change (0.5 mm per month) | Lower mean residual ($p < 0.05$) after change (1.5 mm per month) |
| Positive score | 0.2% of pixels at $p < 0.1$ | 3.7% of pixel at $p < 0.1$ |
| Field significance % of scores | Both positive and negative % within random distribution | Positive % outside random distribution. |

Overall the results suggest that at least for the NSW/VIC region, a decrease in tree cover causes a decrease in rainfall. However, there are several possible complicating factors in the data that require discussion.

## Rainfall variability

Rainfall in Australia is highly variable from year to year, and the time period in this study included a severe drought (*Van Dijk et al., 2013*), as well as the drought breaking years

2010 and 2011. The purpose of using the regression model is to remove the year to year and month to month variability in rainfall and therefore strengthen the land cover signal in the residuals. We used the spatially averaged monthly rainfall timeseries for the larger Murray Darling Basin (Fig. 1) to remove the general variability. However, overall the model does not explain more than 50% (QLD) and 30% (NSW/VIC) of the rainfall variability (Fig. 6), and only around 7% on average is due to the climate drivers (Fig. 7). And while this is consistent with the literature, e.g. *Westra & Sharma (2010)*, this means some variation due to external factors could still be left in the residuals. Rainfall is generally considered a stochastic process (e.g., *Fowler et al., 2005*; *Cowpertwait, Salinger & Mullen, 2009*; *Burton et al., 2010*) and some of the variability could either be a different response to a combination of climate factors (as interactions were not tested in the model), or a non-stationary response to the climate drivers. The severe bushfires in 2003 were also triggered by the extreme drought conditions during the millenium drought (*Van Dijk et al., 2013*). Although the effect of the drought on the overall rainfall quantities has been accounted for in part by the model, a further delayed or cumulative effect of the drought could be feeding into the local land-atmosphere interaction. As a result, the rainfall feedback to the vegetation cover change could be weak under the dry conditions between 2001 and 2009, and this could have affected the result.

The fact that no significant long trend was identified, may not disprove a long term trend in rainfall, as overall time period is fairly short (*Koutsoyiannis, 2006*). Removing the long term trend could mean more pixels in NSW/VIC would indicate a significant step change.

## Vegetation dynamics

The second possible effect is the dynamic nature of the vegetation clearing and recovery, especially for the QLD region. Not only does this refer to a change in the total biomass, but this could also include a change in the species composition as a result of the disturbance.

Although land clearing has occurred at a high rate and broad scale in Queensland (*Department of Science, Information Technology and Innovation, 2017*), the clearing does not have a clear start and end point. QLD has a long history of land clearing. According to the series of SLATS reports on land cover changes in QLD released by the Queensland government, land clearing continued in and around the study region between 1988–2008. Major broad scale and high rate clearings occurred in 1999–2000 and 2002–2004 (Fig. 11). And even though there was a decrease in land clearing post 2005, it is difficult to define a clear cut change in the tree cover in this region.

The specific vegetation class in the Queensland area is also well-known for rapid regrowth and "thickening" in favourable conditions (*Gowen & Bray, 2016*), and this could explain the increase in vegetation cover in 2015 in the QLD area (Fig. 5). This again, could also be related to a change in the vegetation composition further complicating the analysis. In particular the favourable rainfall years of 2010 and 2011 would have boosted regrowth, increasing evapotranspiration and therefore decreasing the effect of the rainfall change.

The different causes of vegetation cover change in these two regions could lead to different post-change characteristics. The trends in EVI in the QLD region are opposite
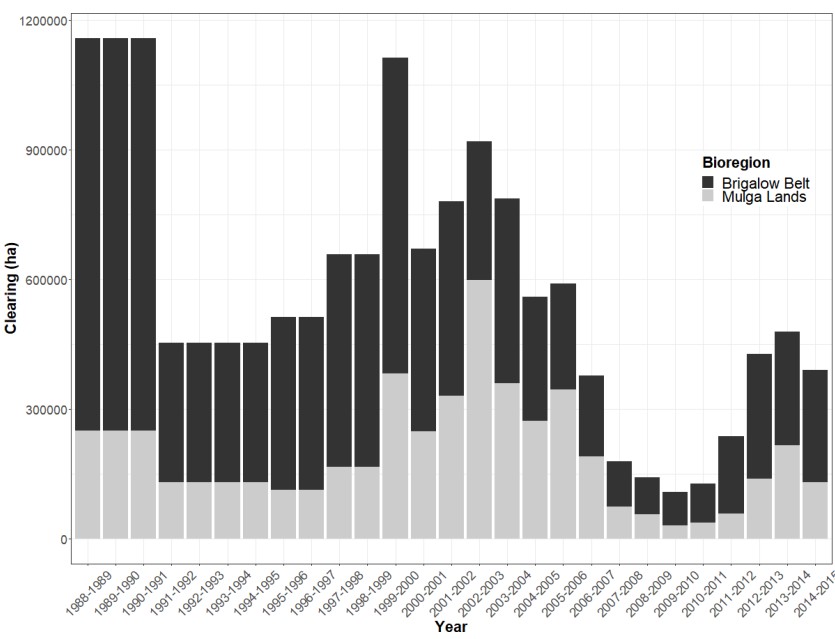

**Figure 11  Woody vegetation clearing rate in QLD for the two major bioregions in the study area.** The data were obtained from SLATS (2017), and clearly indicate a sharp decrease in the clearing rate after 2005.

to the NSW/VIC region, based on the DLCD data. This could also be due to the lower tree density in the QLD region compared to the NSW/VIC region before land surface interventions. In contrast, the significant bushfires (Fig. 2) would have drastically reduced the vegetation cover and recovery was very slow in some areas within the NSW/VIC area. The persistent drought in the 2000s (*Howden, 2012*; *Van Dijk et al., 2013*) delayed the regrowth of trees. Conversely, replacing tree cover with pasture and crops in the QLD area might only have a relatively subtle impact on the EVI.

## Gridded monthly rainfall data

The rainfall data used in this study is gridded. This data set is robust and consistent over a long time series (from 1900 to current) and has a broad national wide coverage which can provide more information spatially (*Jeffrey et al., 2001*; *Tozer, Kiem & Verdon-Kidd, 2009*). However, high cross correlation between pixels, due to the interpolation method in this data set (*Jeffrey et al., 2001*), can also introduce spatial noise. In the step trend test the cross correlation was accounted for. However, some other methods are also available, which could be used to perform a comparative trial. For example, *Narisma et al. (2007)* applied a spatial Gaussian filter on a similar data set and used wavelet analysis to detect a step change in rainfall. High quality station data is another option to test whether the observed spatial pattern in the step trend test results was not due to the gridded data itself. Resampling methods, such as bootstrapping and permutation (used in this study) (*Wilks, 1997*; *Kundzewicz & Robson, 2004*; *Westra, Alexander & Zwiers, 2013*), can also be used to further assess the strength of the significance of results . While the gridded data set is most

useful in regions with sparse rain gauge networks, such as in this study, it can actually reduce information where the rain gauge density is high (*Jones, Wang & Fawcett, 2009*). In the NSW/VIC region, the coverage of rainfall stations is more intensive, but they are mainly located in the valleys. As a result, the interpolated data might be a limited representation of the true local rainfall.

## General approach

Parametric tests are generally more powerful than nonparametric test in detecting a trend, when the data is normally distributed (*Onoz & Bayazit, 2003*; *Kundzewicz & Robson, 2004*). As a non-parametric test, the step trend test has the advantage of being distribution free and having no restriction on missing data (*Hirsch & Gilroy, 1985*). On the other hand, the disadvantages of non-parametric tests, such as being limited to hypothesis testing and being weaker in power, also hold for the step trend test (*Whitley & Ball, 2002*).

Overall, the current study provides a statistical data based approach, building on several lines of evidence to reject the null hypothesis (no step change in rainfall occurs as a result of tree cover loss) at least for the NSW/VIC region. Limited by the available data, the time frame under study had to include a long lasting drought period (*Holper, 2011*; *Van Dijk et al., 2013*). The strong impact of this prolonged drought might have suppressed the land-atmosphere interaction and modified the cause and effect relationship between rainfall and vegetation cover change. This could be one of the reasons that the land cover change effects on the local climate found in other studies (e.g., *Görgen et al., 2006*; *McAlpine et al., 2007*) do not appear in the QLD region.

Future research will include additional areas to further test the approach in this study. However, such areas are not easy to find. There are several requirements for a good study area:

1. It needs to be large enough to capture the effect of landuse on rainfall within the area;
2. It needs to have a drastic enough landuse change to be observable above the rainfall variation;
3. It needs to have a long enough climate data time series both prior and post landuse change.

The obvious locations have always been Amazonia and sub-Saharan Africa, where also most of the modelling studies have taken place (*Kucharski, Zeng & Kalnay, 2013*; *Pitman & Lorenz, 2016*). The problem with the sub-Saharan location is that the land use change is quite long ago and local data is difficult to obtain. The problem with the Amazonian location is that this area is well-known for the feedback and we felt that we could not add much to this work. We propose that it would better to focus on areas affected by bushfires, as we did with the NSW/VIC location. The recent bushfires in California and Colorado could potentially be good locations, but at the moment there is insufficient "post event data". Further work could also focus on identifying an area that has been impacted by land cover change, but does not include a significant drought period.

While the power of the test can be improved with the longer length of the post-intervention period (*Hirsch & Gilroy, 1985*), the dynamic nature of vegetation regrowth in this case study also affects this effect. A better approach might be to build a global study that

investigates multiple locations where drastic landcover changes have taken place, which would also remove some of the climate variability effects due to the larger sample size.

## CONCLUSIONS

In this study, we present a statistical approach to identify the impact of a change in land cover on local rainfall. The results, based on gridded rainfall data found, that a reduction in vegetation cover is likely to have reduced local rainfall for a large area in NSW/VIC affected by bushfires. However, land clearing in QLD was unlikely to have reduced rainfall over the same time period.

Drought may have had a pronounced impact on the land surface condition during the study period, such as leading to significant reduction in vegetation cover and extreme events such as bushfires. The lack of rainfall and associated high temperatures may mask the impact on rainfall of a step change in the vegetation cover. Hence, the signal of Land Cover Change feedback on rainfall is probably weaker under such regional dry conditions, as the impact of Land Cover Change on rainfall is mainly through changes in moisture convergence (*Görgen et al., 2006*; *Pitman & Hesse, 2007*).

### Funding
Chun Xia Liang was supported by an Australian Postgraduate Award for this work. The funders had no role in study design, data collection and analysis, decision to publish, or preparation of the manuscript.

### Grant Disclosures
The following grant information was disclosed by the authors:
Australian Postgraduate Award.

### Competing Interests
The authors declare there are no competing interests.

### Author Contributions
- Chun Xia Liang conceived and designed the experiments, performed the experiments, analyzed the data, prepared figures and/or tables, approved the final draft, initial file organisation and design of repository.
- Floris F. van Ogtrop conceived and designed the experiments, performed the experiments, analyzed the data, contributed reagents/materials/analysis tools, authored or reviewed drafts of the paper, approved the final draft.
- R. Willem Vervoort conceived and designed the experiments, performed the experiments, analyzed the data, contributed reagents/materials/analysis tools, prepared figures and/or tables, authored or reviewed drafts of the paper, approved the final draft.

## Data Availability

The data and code are available at Github via https://github.com/WillemVervoort/RainfallLandcover.

## Supplemental Information

Supplemental information for this article can be found online at http://dx.doi.org/10.7717/peerj.7523#supplemental-information.

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
