# Peer review of "Detecting the impact of land cover change on observed rainfall"

_PeerJ, doi:10.7717/peerj.7523_

## Round 0.1 · original submission · Major Revisions

We now have received three review reports on your manuscript. Based on these review reports, I advise you to further revise your paper.

·

Basic reporting

No comment - very well presented.

Experimental design

No comment

Validity of the findings

No comment

Additional comments

I think the authors should continue to investigate this topic as one of the case studies - the Queensland one - proved probably inconclusive. They should also try and see if they can find instances of increase of land cover and increased rainfall.

Minor points:
Line 50 – problem with El Nin ̃o
Lone 97 – problem with Koppen
Queensland study area – did not have much tree cover initially.
Line 216 – I would have thought Spearman would be more appropriate
Line 306 – below??
Lines 360-361 – something wrong with figure reference

Reviewer 2 ·

Basic reporting

Line 34, maximum temperature at what height? surface temperature?

Line 39, "in both space and time", omit "in"

Line 41, better to add more details/examples about "those complex set of interactions"

Line 48, "On the longer time scale", compare to daily which is not mentioned in previous sentence. This paragraph is comparing local vs. larger scale, not on the time scale

Line 54, should this topic sentence and this paragraph not restricted to "predict Australian rainfall"? but rainfall at any locations?

Line 90, either use "the largest" or "the worst", they mean the same thing

Figure 3, no (a) and (b) were labeled in the graph, or use (left and right).

Table 1, extra space between row "2002: and row "2005"

Experimental design

Line 92, "..due to the drought and cold condition", the reference for that scientific conclusion is from News, a reference from a scientific journal would be more appropriate

Line 113, please include scientific name (Latin) for those tree species.

Line 115, previous sentence authors mentioned about great height, so here better to add details of height. "...20 years to mature (~? m)"

Line 115-116, changing subjects. "Land clearing" is the subject, but not for the rest of the sentence.

Line 125, add more details, t test for the averaged precipitation before (1979-2003) and after (2004-2015) the disturbance? Or move this sentence to the section of "statistical analysis" in below

Line 208, ", this variable was included" what variable? subject changes within a sentence, rewrite

Line 225-231, those paragraphs are results, do authors need them to be here for explaining a further step analysis? or should be moved down to Result section

Line 236-246, those two paragraphs sound like a discussion? should it be moved down to Discussion section?

Line 287-301, should these two paragraphs be combined with each individual analysis below? I don't see any reason to give a summary first and then introducing each analysis

Validity of the findings

Line 363-367, those are not key findings and should be moved to Method section.

Line 377-382, except the first sentence, the rest should be moved down to the discussion section

Line 386, "..compared to the number in QLD region"

Line 394, put the key findings as the subheading text instead of "step trend test" would be better

Line 395-398, those sentences are illustrating how to read the figure, should be put in the figure caption, not here in the Result section

Line 400, what is a "reasonable relationship"? authors should use more statistically sound words to describe the results

Line 404, a difference of what?

Line 411, if authors knew it was difficult to see, why not regraph the data to make it easy to see?

Line 421-429, All those sentences read like Method to me

Line 439 and 440, "strongest evidence" "weak evidence"? could it be more statistically described? like p value or with how much variance explained?

Line 476, the discussion point of "recovery" is worth to expand. Not only the evolution model from Figure 3 which only illustrates the mass/area, the species composition might be changed too. And ties into the evapotranspiration rate.

Line 526, "Possible future work could focus on a non-drought period.." I think research on lands with disturbance(e.g. fire, drought, etc) is also important even though it is hard to study as reported from this study.

Additional comments

This paper uses sophisticated statistical approaches to identify impacts of changes in land cover on rainfall from 1979 to 2015 in two regions in Australia. Results show lands under historical bushfire had significant changes in precipitation, but not for lands with tree clearing. I am sure this paper adds an important value to the community. However, this paper has a very detailed Method section which includes some results and discussion. Some part of the Method section also includes a summary test first and then introducing each analysis. This would cause repetitive sentences. For example, line 303 "As indicated, the step trend test is a modified version of the Mann-Whitney U statistic (Hirsch and Gilroy 1985).", this sentence is repeating line 289 in the summary. I think the summary is not necessary and should be merged to each analysis.
I suggested authors to reorganize the whole paper and keep the sections simple and clear (not overwritten with details not necessary for readers to understand the analysis/paper). Or if authors think those sentences in the Method section are necessary, put them in the appendix, so the main body of this paper can be simplified. I understand authors trying to describe all their trails, thoughts and steps in analyzing those data in this paper. This is my personal opinion. I will let the editor decide the format of this paper.

·

Basic reporting

No comments

Experimental design

No comments

Validity of the findings

No comments

Additional comments

In this paper, Vervoort and his colleagues reported their finding about the impact of land cover change on observed rainfall. The analysis is very comprehensive. And I learned a lot from this manuscript. The authors used statistic approach to analyze observed data, to detect the change of precipitation after forest cleaning and bushfires. They found different influences after two changes. The methods are quite reliable, and conclusions can be trusted. I think this paper has great potential to be a high-cited paper. But the paper still needs to be further improved before publication.

1. Not only the statistically analysis, but also the physically meaning should be further discussed. Theoretically, the change of vegetation will impact on ET, especially in dry seasons, because land cover change will impact on the root zone storage capacity, which provides an important buffer to increase resilience to drought (Gao et al., 2014). This should impact on downwind precipitation.
2. Several important papers, on the impact of land cover on precipitation, are still missed in the reference list (e.g. Keys et al., 20142).

Gao, H. , Hrachowitz, M. , Schymanski, S. J. , Fenicia, F. , Sriwongsitanon, N. , & Savenije, H. H. G. . (2014). Climate controls how ecosystems size the root zone storage capacity at catchment scale. Geophysical Research Letters, 41(22), 7916-7923.
Keys, P. W. , Van, d. E. R. J. , Gordon, L. J. , Hoff, H. , Nikoli, R. , & Savenije, H. H. G. . (2012). Analyzing precipitationsheds to understand the vulnerability of rainfall dependent regions. Biogeosciences, 9(2), 733-746.

---

## Round 0.2 · accepted · Accept

By comparing your revised manuscript with the review comments of the previous round, I decided to accept your paper for publication.

·

Basic reporting

No comment

Experimental design

No comment

Validity of the findings

No comment

Additional comments

Well done with the attention to detail in the revision.

Reviewer 2 ·

Basic reporting

None

Experimental design

None

Validity of the findings

Line 387-406, I think these sentences can be merged with different sections below and the Conclusion section. No need for a separate section at the beginning of the Discussion. But it is fine if authors want to keep it that way.

Additional comments

I really appreciated authors hard working on revising this manuscript. I find this paper is acceptable as it is.